# Development of Antimicrobial Cellulose Nanofiber-Based Films Activated with Nisin for Food Packaging Applications

**DOI:** 10.3390/foods11193051

**Published:** 2022-10-01

**Authors:** Diamante Maresca, Gianluigi Mauriello

**Affiliations:** Department of Agricultural Sciences, University of Naples Federico II, 80055 Portici, Italy

**Keywords:** bacteriocins, food preservation, antimicrobial packaging, plastics, bioplastics, shelf-life

## Abstract

The cellulose nanofiber (CNF) is characterized by the nano-sized (fibers with a diameter between 5 and 20 nm and a length between 2 and 10 μm), flexible and cross-linked structure that confer enhanced mechanical and gas barrier properties to cellulosic fiber-based packaging materials. The purpose of this work was to develop an antimicrobial packaging film by direct mixing nisin with CNF, followed by coating it onto polyethylene (PE), polypropylene (PP), and polylactic acid (PLA) films. The antimicrobial effectiveness of CNF-Nis+PE, CNF-Nis+PP, and CNF-Nis+PLA was investigated both in vitro end in ex vivo tests. In the latter case, challenge test experiments were carried out to investigate the antimicrobial activity of the coupled films of CNF-Nisin+PLA to inhibit the growth of *Listeria innocua* 1770 during the storage of a meat product. The films were active against the indicator microorganisms *Brochothrix thermosphacta* and *Listeria innocua* in in vitro test. Moreover, a reduction in the *Listeria* population of about 1.3 log cycles was observed immediately after the contact (T0) of the active films with hamburgers. Moreover, when the hamburgers were stored in active films, a further reduction of the *Listeria* population of about 1.4 log cycles was registered after 2 days of storage. After this time, even though an increase in *Listeria* load was observed, the trend of the *Listeria* population in hamburgers packed with active films was maintained significantly lower than the meat samples packed with control films during the whole storage period.

## 1. Introduction

Food packaging does not only provide a containment function but also plays a key role in food protection and preservation, convenience, and customer communication strategy [1,2]. Over the last few years, the increase in consumers’ food quality and safety demands has led to a need to develop innovative concepts in the packaging field. In fact, modern packaging systems have been designed to protect packed food, to providing quality and health safety assurance, and to making easier food storage and distribution. In this scenario, the field of active packaging systems is becoming increasingly significant in food technology. Active packaging can be defined as an intelligent or smart system that involves interaction between package or package components and food or internal gas atmosphere and satisfies the consumer demands for minimally processed, high quality, and safe foods [3,4]. Within of subset of active packaging, antimicrobial packaging has received considerable scientific interest leading to extensive research on it. The primary purpose of food antimicrobial packaging is to reduce, inhibit or delay the growth of spoilage and pathogenic microorganisms and thus extend the shelf-life, improve safety, and preserve the quality of packaged products [5]. In the last years, many scientific papers have been published in this field, describing both the antimicrobial compounds and the type of packaging material involved [6,7]. With regard to antimicrobial compounds, organic and phenolic acids, essential oils, anthocyanins, enzymes, bacteriocins, and fungicides are the main substances used to confer antimicrobial properties to food packaging [8]. Among them, bacteriocins, particularly nisin, have been subjected to extensive research in antimicrobial packaging solutions for food biopreservation [9,10]. Nisin, produced by some *Lactococcus lactis* subsp. *lactis* strains, is an antimicrobial peptide recognized as safe by FAO/WHO and Expert Committee on Food Additives, and it was given a generally regarded as safe (GRAS) status for use in meat, dairy products, vegetables, and processed cheese [11]. Nisin was used to activate packaging materials by direct incorporation into the packaging compounds, by coating onto the surface of the packaging, or by film forming using antimicrobial polymers [12,13]. Low-density polyethylene (LWPE), polypropylene (PP), polylactic acid (PLA), sodium caseinate, alginate, chitosan, corn zein, and soy protein were the main plastic and bioplastic materials used to develop antimicrobial packaging films with nisin [14,15,16,17]. However, due to the increasing demand for bio-based packaging materials derived from renewable sources, industry and academic research has been extensively conducted to develop environmentally friendly materials as a sustainable alternative to non-degradable plastics [18,19]. Advances in the nanotechnology field have focused the research attention on nanomaterials derived from cellulose for their common feature characteristics as biodegradability, biocompatibility, non-toxicity, and chemical stability [20,21]. In particular, the cellulose nanofiber (CNF) is characterized by the nano-sized (fibers with a diameter between 5 and 20 nm and a length between 2 and 10 μm), flexible and cross-linked structure that confers enhanced mechanical and gas barrier properties to cellulosic fiber-based packaging materials [5]. Due to these promising characteristics, several studies have reported the use of nano-cellulose-based materials as a coating layer to confer good barrier function towards oxygen and fat [7,22] or as a filler to improve the mechanical and barrier properties as well as the thermal stability of the nanocomposites for potential food packaging applications [5,22,23,24,25]. Moreover, many authors have investigated the antimicrobial efficacy of cellulose-based films when they incorporated the most studied bioactive compounds such as bacteriocins [13,18,26,27,28,29], essential oils [30,31,32], probiotics [33,34], anthocyanins [35,36], and chitosan [19,37,38] for sustainable antimicrobial packaging systems development. However, few investigations have been carried out on cellulose-based materials activation with nisin. In particular, nisin was added to cellulose-based film by (*i*) incorporation into hydroxypropyl methylcellulose and nanocrystalline cellulose [26,27,39,40]; (*ii*) coating onto modified-nanocellulose film surface [18,41,42]; (*iii*) chemical immobilization [28]. Yang et al. [13] developed a bio-based antimicrobial packaging film using nisin and sugarcane bagasse nanocellulose without additional chemical modification. Thus, the purpose of this work was to develop an antimicrobial packaging film by direct mixing nisin with CNF water suspension, followed by coating onto polyethylene (PE), polypropylene (PP), and polylactic acid (PLA) films. In addition, the antimicrobial effectiveness of CNF-Nisin+PE, CNF-Nisin+PP, and CNF-Nisin+PLA was investigated both in vitro and in ex vivo tests. In the latter case, challenge test experiments were carried out to investigate the antimicrobial activity of the coupled films of CNF-Nisin+PLA to inhibit the growth of *Listeria innocua* 1770 throughout the storage time of a meat product.

## 2. Materials and Methods

### 2.1. Bacterial Strains and Culture Conditions

*Brochothrix thermosphacta* 7R1 and *Listeria innocua* 1770 were used as indicator strains in the nisin bioactivity assay. They have been provided by Department of Agricultural Sciences, Division of Microbiology, University of Naples Federico II, and were previously identified by 16S rRNA gene sequencing [43,44,45]. *B. thermosphacta* 7R1 and *L. innocua* 1770 were routinely cultured in Tryptone Soya Broth (TSB, Oxoid, Milan, Italy) supplemented with 5 g/L Yeast Extract Powder (Oxoid S.p.A., Milan, Italy) at 20 °C and 30 °C, respectively.

### 2.2. Preparation of Nisin Solution (Nis)

The nisin solution (Nis) was obtained and tested as described by Maresca et al. [46] with some modifications. Briefly, 0.5 g of Nisin (Sigma, 2.5% of nisin) were dissolved in 10 mL of 0.02 mol/L HCl (Merck, Vimodrone, Italy) and centrifuged at 6500× *g* for 15 min. Pellet was treated in the someway twice, and all supernatants from each centrifugation process were mixed in a single solution representing the Nis. It was routinely stored at 4 °C for further experiments. Its antimicrobial activity was tested against *B. thermosphacta* 7R1 and *L. innocua* 1770 by agar diffusion and critical dilution assay as described by Villani et al. [47]. Briefly, serial two-fold dilutions of bacteriocin in 0.02 mol/L HCl were prepared, and 10 μL of each dilution were spotted onto TSA soft agar (TSB with the addition of 7.5 g/L agar and 5 g/L Yeast Extract Powder) seeded with 25 μL/mL of an overnight culture of each indicator strain. After overnight incubation at 20 °C for *B. thermosphacta* and 30 °C for *L. innocua*, the arbitrary units per ml (AU/mL) of Nis were calculated as the reciprocal value of the highest dilution that showed clear growth inhibition of the indicator strain.

### 2.3. Preparation of CNF-Based Films

Different quantities of CNF were dissolved in a Nis water solution to reach a CNF concentration of 1.0, 1.5, 2.0, 2.5, and 3.0% (*w*/*v*) and an antimicrobial activity of 640 AU/mL and 1280 AU/mL against *B. thermosphacta* and *L. innocua*, respectively. Control mixtures were prepared at the same CNF concentrations in water without nisin. The mixtures were stirred at 300 rpm for 15 min to favor the homogeneous CNF distribution and the CNF-nisin blend formation. Afterward, 50 mL of each mixture was poured into 90 mm diameter glass Petri dishes, previously loaded on the bottom with circle sheets of 90 mm in diameter of polyethylene (PE), polypropylene (PP), or polylactic acid (PLA). Dishes were incubated at 60 °C for 16 h for water evaporation and the CNF-based films formation.

### 2.4. Antimicrobial Activity Test

Only the films obtained by using 2.5% and 3.0% CNF mixtures were tested here. Coupled films with nisin were labeled CNF-Nis+PE, CNF-Nis+PP, and CNF-Nis+PLA, according to the plastic material used. Control coupled films were labeled CNF/PE, CNF/PP, and CNF/PLA. The antimicrobial activity was evaluated immediately post films manufacturing (T_0_) and after storage for 7 days (T_7_) at 4 °C. Briefly, pieces of active and control films, as well as of plastic material alone, were placed onto the surface of TSA soft agar plate seeded with 2.5% of an overnight culture of each indicator strain. After plates incubation (20 °C for *B. thermosphacta* and 30 °C for *L. innocua*), the antimicrobial activity was evaluated by observing a growth inhibition zone underneath and around the active film samples [14].

### 2.5. Scanning Electron Microscopy (SEM) Analysis

The surface morphology of films was examined using an FEI Quanta 200 FEG scanning electron microscope (SEM) (Thermo Fisher Scientific Inc., Waltham, MA, USA) in high vacuum mode. Before observations, samples were mounted onto stubs by means of carbon adhesive discs (Agar Scientific Ltd, Stansted Mountfitchet, UK) and sputter coated with a 5–10 nm thick Au-Pd layer. All samples were analyzed at 10 kV acceleration voltage using a secondary electron detector. SEM images were recorded at magnifications of 5000×, 10000×, and 25000×.

### 2.6. Challenge Test

Challenge test experiments have been carried out to validate the CNF-Nis+PLA film for functioning in food packaging applications. Fresh minced meat (provided from local market and immediately transferred to the laboratory in refrigerated conditions) was artificially contaminated with 80 mL of an *L. innocua* 1770 suspension (2.0 × 10^6^ CFU/mL) to obtain food contamination of about 8.0 × 10^4^ CFU/g. After manual homogenization of the meet, hamburgers were formed (approximately 70 g each) and covered on both sides with the CNF-Nis+PLA film at 2.5% of CNF. Hamburgers covered with CNF/PLA film were included in the analysis as control. All samples were stored at 4 °C for a week. Bacterial loads of *L. innocua* 1770 were monitored by viable plate count technique immediately after meat contamination (unpacked meat) (T0_0_), immediately after setting hamburgers packaging (T0), and after 1, 2, 3, 6, 7, and 8 storage days. At each sampling time, aliquots (20 g) were taken and homogenized in 180 mL (*w*/*v*) of quarter strength Ringer’s solution (Oxoid, Milan, Italy). Subsequently, ten-fold serial dilutions were carried out, and *L. innocua* 1770 was enumerated on Listeria Chromogenic Agar Base according to Ottaviani and Agosti (ALOA, Biolife, Milan, Italy). Furthermore, aliquots of 0.1 mL of the appropriate dilutions were also spread onto PCA (Plate Count Agar, Oxoid) plates for the enumeration of total mesophilic aerobic bacteria. All plates were incubated at 37 °C for 24–48 h. The experiments were performed in triplicate, and the results were expressed as Log CFU/g of product.

### 2.7. Data Analysis

Analyses were carried out in triplicate, and all data were reported as mean ± standard deviation. Two-way ANOVA tests and t-test analyses (Microsoft Excel for Mac version 11.5, Microsoft Corporation, Redmond, WA, USA) were performed to evaluate significant differences (*p* < 0.05) between means.

## 3. Results

### 3.1. CNF-Based Films Manufacturing

All the films produced with a 2.5% and 3% mixture of CNF were found to be easily peeled off to petri dish, and they showed a uniform interfacial adhesion between PE, PP, PLA layer, and the CNF (control) or CNF-Nis (active) layer. On the contrary, films containing 1, 1.5, and 2% of CNF showed poor adhesion to the plastic layer and a non-continuous surface with crack zones. Based on visual observations, both active and control coupled films obtained with 2.5% and 3.0% of CNF and on all different polymers were not transparent, and they showed completely different surface morphology.

In Figure 1 is represented, as an example, the coupled activated film obtained by using a 2.5% CNF mixture and PLA. The surface of the control film (Figure 1, panel A) appeared smooth and continuous layer without cracks zones, while the surface of the activated film (Figure 1, panel B) appeared wrinkled with marked ice-crystal-like features.

### 3.2. Antimicrobial Efficacy of CNF-Based Films in Agar Inhibition Assay

The antimicrobial activity of all films with nisin at T_0_ and T_7_ was highlighted by a clear growth inhibition zone of the test microorganism *B. thermosphacta* 7R1 and *L. innocua* 1770, beyond the perimeter of the samples and underneath, after removing the film from the agar. The films at 2.5 and 3.0% CNF did not show significant differences in antimicrobial activity according to the dimension, homogeneity, and clearness of the growth inhibition areas (data not shown). In Figure 2 is depicted, as an example, the result of the antimicrobial activity test of the films prepared by using 2.5% CNF after their preparation (T_0_) against *B. thermosphacta* TR1. On the left side are represented the Petri dishes before removing the film from the agar, and the growth inhibition area is clear beyond the films with nisin (sections A, D, and G). No other film led to the inhibition of the growth (all other sections). On the right side are represented the Petri dishes after removing the film, and the growth inhibition zones are very clear underneath the film samples with nisin (sections A1, D1, and G1). On the contrary, a lack of antimicrobial activity was confirmed for the control coupled films and polymers alone (all other sections).

### 3.3. Surface Characterization of CNF-Based Films

In Figure 3 are depicted the SEM images at different magnifications of the surface morphology of the CNF/PLA control film (Figure 3 panels A, B, and C) and CNF-Nis+PLA active film (Figure 3 panels A1, B1, and C1), both at 2.5% of CNF. The surface of CNF/PLA film is characterized by a continuous network of cellulose nanofibers with few crack zones between fibers forming a compact coating matrix. As shown in Figure 2 (panels A1, B1, and C1), the surface of the CNF-Nis+PLA active film appears to have a different internal structure compared to the control film. In particular, the CNFs network appears less homogeneous with pores formation in some areas.

### 3.4. Antibacterial Effect of CNF-PLA Films in a Food System

Results of the viable counts of *L. innocua* in hamburgers packed with active (CNF-Nis+PLA) and control (CNF/PLA) coupled films along 8 days of storage at 4 °C are reported in Figure 4. A reduction in *Listeria* population, of about 1.3 log cycles, was observed immediately after the contact (T0) between hamburgers and the active films compared with the control. Moreover, when the hamburgers were stored in active films, a further reduction of *Listeria* population, of about 1.4 log, was registered after 2 days (T2) of storage. After this time, even though an increased *Listeria* load was observed, the trend of the *Listeria* population in hamburgers packed with active films was maintained significantly lower than the meat samples packed with control films during the whole storage period (Figure 4). In Figure 5 are shown the results of total viable mesophilic counts in hamburgers packed and stored in the same conditions. As can be seen, in hamburgers packed with control films, the number of total bacteria increased from 5.5 to 6.6 log CFU/g after 8 days of storage. On the contrary, the total bacteria count of the hamburgers packed with the active film was significantly lower along 8 days of storage compared to the control.

## 4. Discussion

In this work, we attempted to combine the functionalization of the CNF with nisin and the casting of a CNF-nisin solution onto PP, PE, and PLA films to develop coupled antimicrobial CNF-based films for food packaging applications. The novelty of this study lies in the solution cast method used to obtain active coupled films without any chemical or enzymatic pre-treatment to CNF as well as to plastic or bio-plastic films. Our results suggested that the coupled film formation depended greatly on CNF concentration in the mixtures. In fact, the best results were obtained only using the higher CNF content of 2.5% and 3.0%. In particular, these films (both active and control) exhibited a compact and homogeneous surface structure without crack zones as well as a uniform adhesion between PE, PP, and PLA layers and the CNF alone (control) or CNF-Nis layer (treated). Firstly, this result suggested that a minimum of CNF content is probably necessary to promote the interactions among nanofibers and the formation of a continuous fibrillar network, which led to no collapse onto film surfaces during the drying. Notably, no previous research investigated the effect of different CNF concentrations in combination with nisin on surface film morphology. However, some authors have provided some information that could support the relationship between the different CNF concentrations, the surface structure of the films, and the physico-mechanical properties of CNF films. Azeredo et al. [48] observed that the mechanical properties of the mango puree-CNF edible films were improved at higher CNF concentrations. Pizzaro et al. [49] reported that the increase in the CNF concentration (from 1 to 5% *w*/*w*) resulted in a decrease in the water vapor permeability of edible films. Finally, Lapuz et al. [50] proved that as the concentration of CNF increased from 0.2% to 1.0%, the surface of the edible film appeared smoother without breaking zones. Additionally, the same authors observed an improved tensile strength of the films when the CNF concentration was increased. Thus, the surface integrity and physico-mechanical properties of CNF films can be dependent on the CNF content. In fact, all mentioned authors suggested that a high CNF content can lead to establishing a close fiber-to-fiber contact by hydrogen bond formation to form a rigid CNF network structure with very interesting resistance behavior. Therefore, the non-continuous surface with several crack zones observed at low CNF concentration can be explained by the bonds weakening or even debonding resulting in less integrity of CNF structure. In our study, the addition of nisin into the CNF solution remarkably changed the surface appearance of the active films developed. Compared to control, the surface of the active coupled films appeared highly rough with ice-crystal-like features. These characteristic surface structures have never been reported in previous studies. Saini et al. [18] noticed only an apparent change in surface color, from white to brown, after nisin grafting on the CNF films. Similar findings were obtained by Yang et al. [13]. These authors noticed a uniform and smooth surface of sugarcane bagasse-CNF-nisin hybrid films and an apparent change in surface color compared to pure CNF films. Moreover, studies based on microscopic investigation of hybrid CNF-nisin films by scanning electron microscope (SEM) observed a smooth and homogeneous surface [27] or a rough and porous surface [40] compared to CNF films without nisin. However, similar crystallized structures were observed on the surface topography of nisin-polyethylene films when they were characterized by the atomic force microscopy technique [51]. According to the authors, these crystallized structures could be attributed to agglomerates formed by ethylenediaminetetraacetic acid (EDTA) added as a chelating agent to the nisin solution. In our case, the nisin solution was added to FFSs without EDTA. Based on our observation, we assumed that the ice-crystal-like structures could be due to salt crystals (sodium chloride salt present in Nisin powder formulation) formation during the drying process. The salt crystal growth phenomena during evaporation of a NaCl aqueous solution have been suggested before [52,53]. In these studies, dendrite structures of NaCl similar to snowflakes were generated on a glass surface by a simple drying of a NaCl solution. However, despite the biological similarity with our results, the surface structures of active coupled CNF-based films are to be further investigated. Microscopic investigation of CNF/PLA control film by SEM analysis confirmed the close contact between CNFs, resulting in a homogeneous structure. On the other hand, the continuous and non-porous surface structures of the CNFs-based films have been demonstrated by SEM analysis in previous research [7,27,54,55]. On the contrary, to the best of our knowledge, our study is the first to analyze the surface morphology of coupled CNF-Nis+PLA active film where the CNF matrix was used as a carrier for the bacteriocin. Moreover, in our study, coupled CNF-based films obtained by using the greatest CNF content also showed a perfect adhesion between plastic (PP, PE) or bio-plastic (PLA) films and CNF-based layers (with or without nisin). To the best of our knowledge, our study is the first to develop active coupled films by casting and drying CNF-nisin solutions directly onto PP and PLA films. Similarly, Yang et al. [13] developed a CNFs/nisin film used as a liner of low-density polyethylene (LDPE) plastic packaging for ready-to-eat ham. Grower et al. [41] prepared nisin-methylcellulose (MC) and nisin-hydroxypropyl methylcellulose (HPMC) solutions coated onto low-density polyethylene (LDPE) films. However, despite the different nature of the cellulose matrix (MC and HPMC), antimicrobial cellulose-based film manufacturing is needed for plasticizer use (glycol 400). Salmieri et al. [42] primarily developed a PLA-CNF nanocomposite film and then functionalized it with nisin by the adsorption coating method. In literature, it is well known the use of CNF as reinforcement in various polymers, including PP, PE, and PLA, to develop composites or bio-composites [56,57,58,59,60,61]. However, in these studies, the CNF polymer composites manufacturing needed specific techniques (extrusion, solution casting, compression molding, and melt spinning) and processing conditions (coupling agents or surfactants utilized, chemical modifications onto CNF surface) to facilitate the uniform distribution of CNF in polymer or bio-polymer matrix [62,63]. As a matter of fact, the key challenge is to obtain compatibility between the hydrophobic polymer matrices and hydrophilic CNF. In our study, we found the perfect adhesion between polymer films and the CNF layer.

Antimicrobial activity of developed CNF-Nis+PE, CNF-Nis+PP, and CNF-Nis+PLA coupled films was observed by agar inhibition assay against both indicator strains, *B. thermosphacta,* and *L. innocua.* These bacteria are actually the main used microbial targets to study the antimicrobial effectiveness of film packaging containing nisin. The uniform and confined inhibition zones suggested a homogeneous dispersion of Nis in the CNF matrix such that it did not diffuse irregularly into the agar media. The compatibility between nisin and CNF has been demonstrated in several investigations. In particular, the chemical interactions between the compounds in the CNF-nisin films have been investigated by SEM, and Fourier transform infrared (FTIR) analysis [13,18,27,39,42]. The surface of the CNF network has a large amount of negatively charged carboxyl groups, while nisin is positively charged at a pH below its isoelectric point [64]. Therefore, when nisin is dispersed through the CNF matrix, electrostatic interactions can occur between positively charged nisin and negatively charged CNFs such that nisin is embedded in the nanofiber network through the formation of hydrogen bondings. According to Lu et al. [64], the close fiber-to-fiber sites are replaced with fiber-nisin-fiber sites through electrostatic interactions. Moreover, all active coupled films showed to preserve the antimicrobial activity for 8 days under refrigerated storage conditions. It is well known that nisin shows its antimicrobial activity against *B. thermosphacta* and *L. innocua*. However, to the best of our knowledge, the present study gives the first experimental evidence of the long-term antimicrobial effectiveness of coupled CNF-based films containing the bacteriocin towards the mentioned indicator strains. In research conducted by Sebti et al. [39], the cross-linked HPMC-nisin films were completely inactive against *Micrococcus luteus* in an agar inhibition assay. This result was due to possible nisin retention via chemical bonds with HPMC or other film components during the film forming process. Instead, Imran et al. [26] developed an HPMC-nisin film that showed antimicrobial activity against different strains belonging to *Listeria monocytogenes*, *Staphylococcus aureus, Bacillus cereus, Bacillus subtilis,* and *Enterococcus facium* species. Finally, the nisin-distarch phosphate-nanocrystalline cellulose films, manufactured by Sun et al. [40], proved in vitro to be effective in inhibiting the growth of *S. aureus* and *E. coli*. In order to validate an antimicrobial film for food packaging application, the antimicrobial properties of active coupled films were also quantitatively investigated by challenge test experiments. During the application study on hamburgers, chosen as a real food system, the CNF-Nis+PLA coupled films showed their effectiveness in inhibiting and controlling the growth of both *L. innocua* and total mesophilic aerobic bacteria compared to control films. The contact between the meet sample and active coupled films caused an immediate reduction in *L. innocua* population. Moreover, viable counts of *L. innocua* were lower by 1.4 log CFU/g at 2 days of storage. After this time, although an increased *Listeria* load was registered, the Nis+CNF/PLA satisfactorily assured the control of the *Listeria* population until the end of the storage compared to control films. The overall trend in *Listeria* viable counts may depend on nisin release and action mechanisms in food matrix. During the first 48 h of storage, the significant decrease in bacterial population could suggest a fast nisin release allowing an immediate bacteriocin action towards the indicator strain. After this time, the gradual *Listeria* growth could be explained by (*i*) a decreased bacteriocin concentration on the film surface, (*ii*) a slower nisin release in the food system, and/or (*iii*) a bacteriocin inactivation by food components (i.e., proteins and lipids) resulting in its reduced availability to act against the target microorganism. Our results do not agree with the data provided by Yang et al. [13]. In this study, the CNFs/nisin hybrid films used as a liner of an LDPE plastic package for ready-to-eat ham showed total inhibition of *L. monocytogenes* during 7 days of storage at 4 °C. Similarly, in Salmieri et al. [42] experiments, PLA-cellulose nanocrystal films containing nisin showed a significant reduction of *L. monocytogenes* in ham from day 1 and a total inhibition from day 3. Instead, when frankfurters were packed with bacterial cellulose films containing nisin, the *L. monocytogenes* population significantly decreased after 2 days of storage and then remained constant until the end of the experiment [65]. However, Hassan et al. [27] carried out experiments based on nisin release from nanofibrillated cellulose films and proved evidence than can support our results. The authors observed a fast diffusion of nisin during the first 6 h resulting in complete inhibition of *S. aureus* over this time. Subsequently, a gradual recovery of bacterial survivors was observed till 72 h due to the slower diffusion and nisin inactivation during this second period. In our study, the antimicrobial effect of Nis+CNF/PLA on the total mesophilic aerobic bacteria population in hamburgers was also evaluated. The initial total bacteria load was approximately 5.5 log CFU/g, and a continuous increase up to 6.6 log CFU/g was observed after 8 days of storage in control coupled films. On the contrary, the total bacteria count of the hamburgers packed with the active coupled film was significantly lower along 8 days of storage compared to the control. Our findings suggested that CNF-Nis+PLA films possessed the potential for controlling the growth of autochthonous spoilage microbiota of the meat product; therefore, they could be promising and innovative antimicrobial packaging systems to enhance the safety and extend the shelf life of the food.

## 5. Conclusions

CNF showed to be a promising material to enhance the functional properties of food packaging. Indeed, it showed the ability to capture the nisin and make it available to work as an antimicrobial agent when in contact with pathogen and spoilage microorganisms, both in vitro and in ex vivo challenge tests. Moreover, our findings showed that natural autochthonous meat microbiota was negatively affected as well by the contact of food packaging obtained by using a nisin-activated CNF layer. Remarkably, the CNF layer (with or without nisin) showed to be compatible with polyolefin plastics, such as PE and PP, and bioplastics, such as PLA, to obtain multilayer packaging for the food industry application.

## Figures and Tables

**Figure 1 foods-11-03051-f001:**
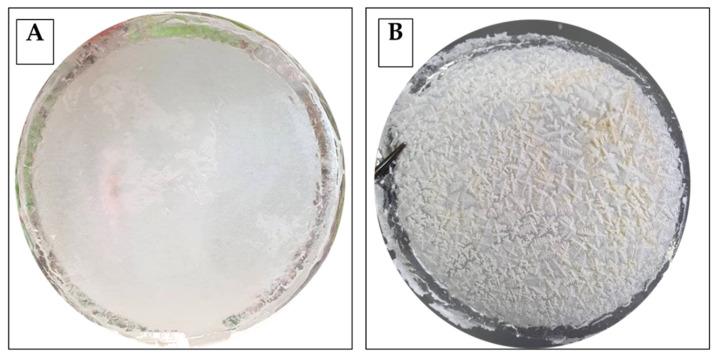
Films obtained by using 2.5% CNF mixture and PLA. Control film (**panel A**); nisin-activated film (**panel B**).

**Figure 2 foods-11-03051-f002:**
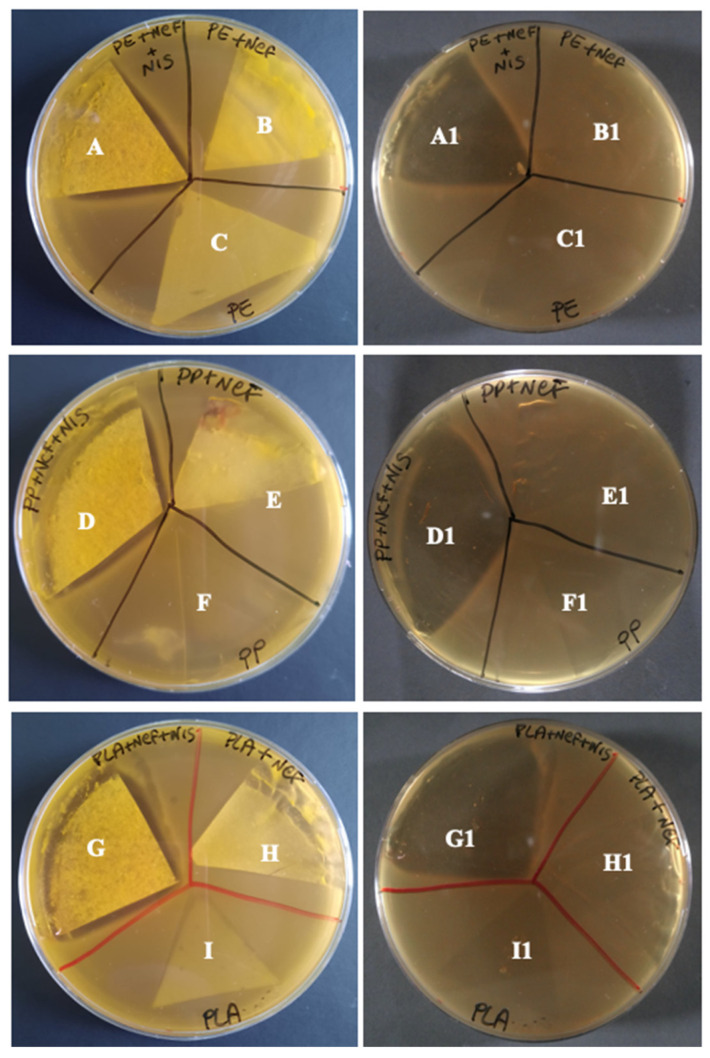
Coupled films and polymers alone in inhibition agar test against *Brochothrix thermosphacta* 7R1 before (**left side**) and after (**right side**) removing the samples from the agar. Segments A and A1, CNF-Nis+PE; segments B and B1, CNF/PE; segments C and C1, PE; segments D and D1, CNF-Nis+PP; segments E and E1, CNF/PP; segments F and F1, PP; segments G and G1, CNF-Nis+PLA; segments H and H1, CNF/PLA; segments I and I1, PLA.

**Figure 3 foods-11-03051-f003:**
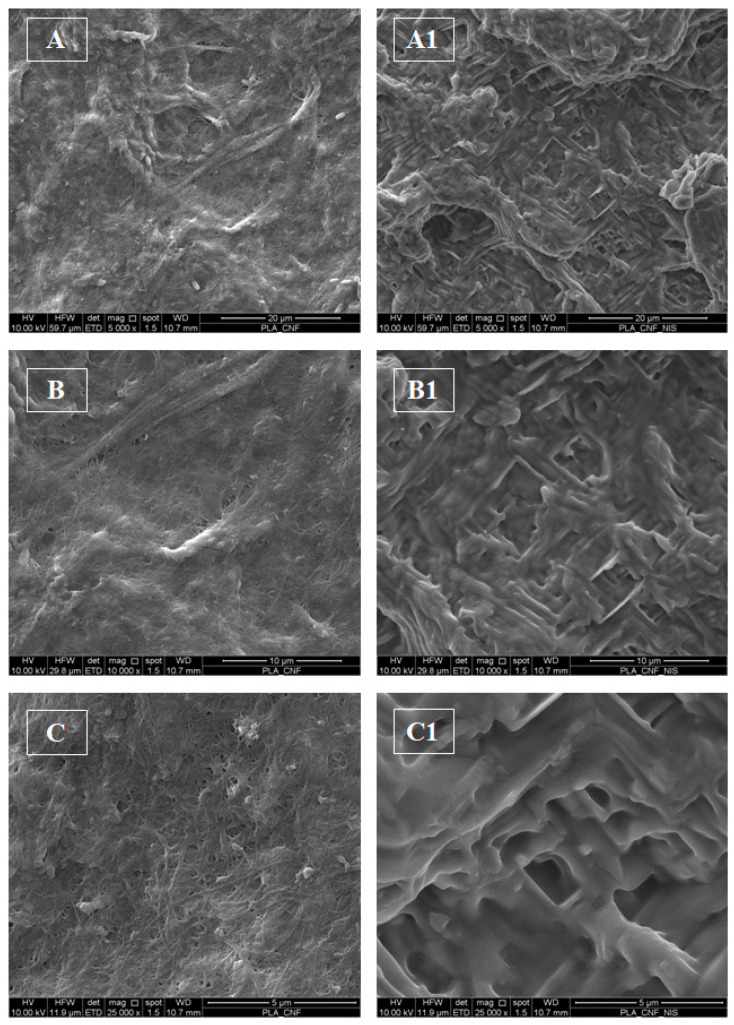
SEM images of CNF/PLA (**left side**) and CNF-Nis+PLA (**right side**) film at 2.5% CNF. (**A**–**C**): CNF/PLA at 500×, 1000×, and 25000× magnifications, respectively; (**A1**–**C1**): CNF-Nis+PLA at 500×, 1000×, and 25000× magnifications, respectively.

**Figure 4 foods-11-03051-f004:**
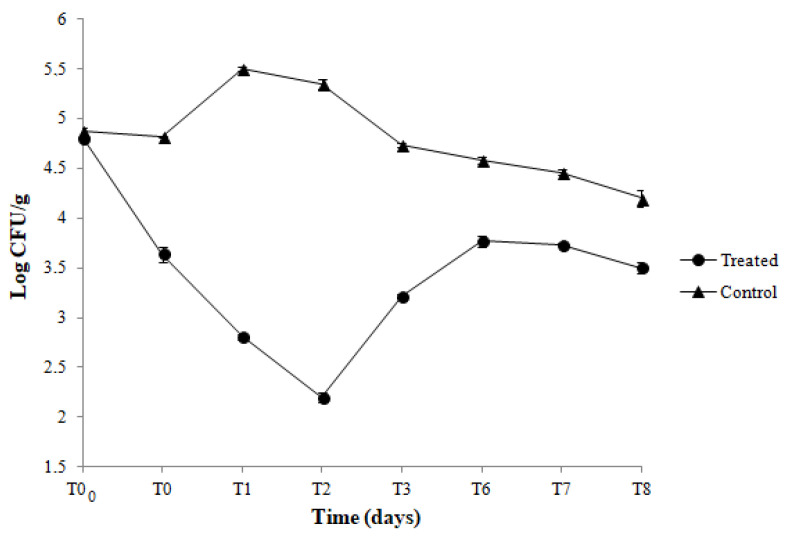
Viable counts of *L. innocua* population during the storage at 4 °C of hamburgers packed with CNF-Nis+PLA active films (treated) and CNF/PLA control films (control).

**Figure 5 foods-11-03051-f005:**
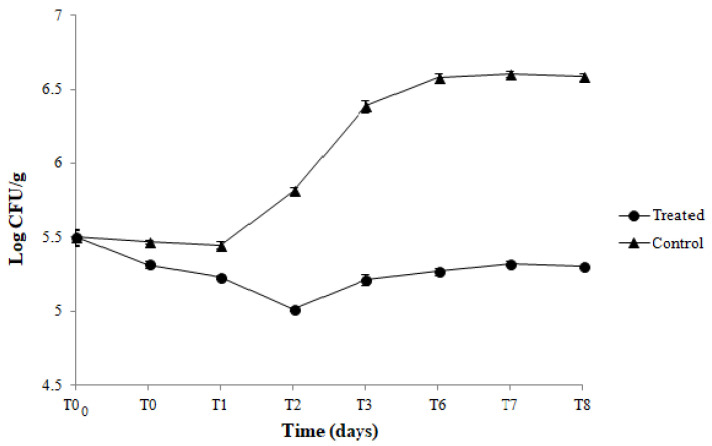
Total viable counts of mesophilic aerobic bacteria during the storage at 4 °C of hamburgers packed with CNF-Nis+PLA active films (treated) and CNF/PLA control films (control).

## Data Availability

Data is contained within the article.

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
