# Peer review of "Development of Antimicrobial Cellulose Nanofiber-Based Films Activated with Nisin for Food Packaging Applications"

_foods, 2022, doi:10.3390/foods11193051_

Round 1
Reviewer 1 Report
I think the study is somehow interesting. However, it can be suggested revisions below.
1. The language is hard to follow which should be polished.
2. The figures as well as some flowchart should be added to make the study easier to understand.
3. The literature part is very poor, some very strong relevant articles are not cited and commented, just to mention some
2021, Active cellulose acetate-carvacrol films: Antibacterial, physical and thermal properties by Denise Adamoli Laroque
2022, Cellulose-based absorbent pad loaded with Carum copticum essential oil for shelf life extension of refrigerated chicken meat by Guangfa Liu
2021, Preparation and characterization of whey protein isolate/chitosan/microcrystalline cellulose composite films by Xinyu Zhai
2021, Active packaging based on cellulose trays coated with layered double hydroxide as nano-carrier of parahydroxybenzoate: Application to fresh-cut iceberg lettuce by Giuliana Gorrasi
Author Response
Q. The language is hard to follow which should be polished.
R. We revised the language.
Q. The figures as well as some flowchart should be added to make the study easier to understand.
R. We clarified in the text to make easier for the reader the logic sequence of the study.
Q. The literature part is very poor, some very strong relevant articles are not cited and commented, just to mention some.
R. We followed your suggestion to improve the literature citation.
Reviewer 2 Report
1. The authors claim that the best result was achieved with a CNF content of 2.5% w/v. However, there is no data about the influence of higher content of CNF on properties of film. It will be useful to study antimicrobial properties of proposed packing films with higher content of CNF.
2. A significant part of the discussion of the results is devoted to the surface morphology of the obtained films. However, it is necessary to conduct a microscopic study of the surface of the film for a deeper understanding of the interaction of the components of the film and its constituents.
Author Response
Q. The authors claim that the best result was achieved with a CNF content of 2.5% w/v. However, there is no data about the influence of higher content of CNF on properties of film. It will be useful to study antimicrobial properties of proposed packing films with higher content of CNF.
R. Thank for your observation. We did preliminary experiments also with 3% of CNF but we didn't report in the first version of the paper because we didn't observe significan differences between 2.5 and 3%. However, we have reported now in the revised version.
Q. A significant part of the discussion of the results is devoted to the surface morphology of the obtained films. However, it is necessary to conduct a microscopic study of the surface of the film for a deeper understanding of the interaction of the components of the film and its constituents.
R. We followed your suggestion and analysed the films by SEM.
Reviewer 3 Report
Active packaging is important for the sustainable development. Biomass are promising materials for the design of biodegradable active packaging. The topic of this manuscript is of broad interest to the readers and the experiments are well designed. However, major revision is required.
1. “plastics” and “bioplastics” in the Keywords are suggested to be replaced one by “nisin”.
2. Expressions like “Recently, [22] developed…” in line 72 need to be revised.
3. The authors need to highlight what are the advantages of the composite films prepared in this work over antimicrobial packaging films?
4. Renewable sources are promising precursors for biodegradable active packaging. Many researches have been done and more references are suggested to be cited, for example Development and Characterization of Food Packaging Bioplastic Film from Cocoa Pod Husk Cellulose Incorporated with Sugarcane Bagasse Fibre; Packaging and degradability properties of polyvinyl alcohol/gelatin nanocomposite films filled water hyacinth cellulose nanocrystals; Electrospun Functional Materials toward Food Packaging Applications: A Review.
5. More characterizations are suggested to be added. For example, SEM images of films are required to show the microstructure of films.
6. Mechanical strength is an important issue need to be considered for packing films. So data on mechanical strength are suggested to be added.
7. Most of the references are too old. More references published recently are suggested to be cited.
Author Response
Q. “plastics” and “bioplastics” in the Keywords are suggested to be replaced one by “nisin”
R. The term "nisin" is in the title and usually journals don't prefer repetitions in title and key words.
Q. Expressions like “Recently, [22] developed…” in line 72 need to be revised.
R. Done.
Q. The authors need to highlight what are the advantages of the composite films prepared in this work over antimicrobial packaging films?
R. We clarify the advantages of CNF implementation in the Introduction.
Q. Renewable sources are promising precursors for biodegradable active packaging. Many researches have been done and more references are suggested to be cited, for example Development and Characterization of Food Packaging Bioplastic Film from Cocoa Pod Husk Cellulose Incorporated with Sugarcane Bagasse Fibre; Packaging and degradability properties of polyvinyl alcohol/gelatin nanocomposite films filled water hyacinth cellulose nanocrystals; Electrospun Functional Materials toward Food Packaging Applications: A Review.
R. We cited some papers and reviews on the topic (references n. 2, 5, 6, 13 .....).
Q. More characterizations are suggested to be added. For example, SEM images of films are required to show the microstructure of films.
R. Done
Q. Mechanical strength is an important issue need to be considered for packing films. So data on mechanical strength are suggested to be added.
R. CNF can modify the mechanical strenght when they are used as fillers (we clrarify in the Introduction). In our case they are used as coating on some polymers having definitely higher mechanical performances, so CNF cannot affect them.
Q. Most of the references are too old. More references published recently are suggested to be cited.
R. Done.
Round 2
Reviewer 3 Report
The manuscript is well revised according to the comments and could be accepted.